

# 1 Forecasting auroras from regional and global magnetic

# 2 field measurements

**K. Kauristie[1], M. Myllys[2], N. Partamies[3], A. Viljanen[1], P. Peitso[1,4], L. Juusola[1],**
**S. Ahmadzai[2], V. Singh[4], R. Keil[5], U. Martinez[6], A. Luginin[5],**
**A. Glover[5], V. Navarro[5] , and T. Raita[7].**
[1]{Finnish Meteorological Institute, Helsinki, Finland}
[2]{University of Helsinki, Finland}
[3]{The University Centre in Svalbard, Norway}
[4]{Aalto University, Espoo, Finland}
[5]{European Space Agency, ESOC, Darmstadt, Germany}
[6]{etamax space GmbH, Darmstadt, Germany}
[7]{Sodankylä Geophysical Observatory, University of Oulu, Finland}
Correspondence to: K. Kauristie (kirsti.kauristie@fmi.fi)
**Abstract**
We present a concept for a Regional Auroral Forecast service (RAF), which uses near-real-
time data from the IMAGE network of ground-based magnetometers operated in Northern
Fennoscandia. Performance of RAF is demonstrated in a case study with auroral recordings
from the Sodankylä research station. RAF is based on archived National Oceanic and
Atmospheric Administration (NOAA) space weather alerts and regional magnetic field
recordings (years 2002-2012). The archives are used to create a set of conditional
probabilities, which tell the service user when the probability to see auroras exceeds the
average conditions in Fennoscandia during the coming 0-12 hours. Favourable conditions for
auroral displays are associated with ground magnetic field time derivative values ($d\mathbf{B}/dt$)
exceeding certain latitude dependent threshold values. Our statistical analyses reveal that the



probabilities to record d**B**/dt exceeding the thresholds stay below 50% after NOAA alerts on
X-ray bursts or on energetic particle flux enhancements. Therefore, those alerts are not very
useful for auroral forecasts, if we want to keep the number of false alarms low.  However,
NOAA alerts on global geomagnetic storms (characterized with Kp values >4) enable
probability estimates of >50% with lead times of 3-12 hours. RAF forecasts thus rely heavily
on the well-known fact that bright auroras appear during geomagnetic storms. The additional
new piece of information which RAF brings to the previous picture is the knowledge on
typical storm durations at different latitudes. For example, the service users south of the Artic
Circle will learn that after a NOAA ALTK06 issuance in night, auroral spotting should be
done within 12 hours after the alert, while at higher latitudes conditions can remain
favourable during the next night.
## 1   Introduction
Space weather is the physical and phenomenological state of natural space environments. The
associated discipline aims, through observations, monitoring, analysis and modeling, at
understanding and predicting the state of the Sun, the interplanetary and planetary
environments, and the solar and non-solar driven perturbations that affect them, and also at
forecasting and nowcasting the potential impacts on biological and technological systems
(Lilensten et al., 2008). Auroras are harmless, fascinating feature of ionospheric space
weather. They are an important factor in the business of nature tourism in polar areas. In this
context there is a demand to get auroral forecast with long lead times - hours, days or even
weeks.
The original energy source for space weather phenomena is the Sun, which emits a wide
spectrum of electromagnetic waves and a continuous flow of charged particles (solar wind) to
its surroundings. Rapid variations in space weather conditions (space weather storms) are
associated with large scale dynamic phenomena (coronal holes, flares and mass ejections)
taking place in the solar atmosphere (corona). The first signs of solar eruptions are X-ray
flares and EUV and radio wave bursts which reach the Earth surroundings with ~8 min delay
after their initiation. The next sign are enhancements in energetic particle fluxes as observed
e.g. at the geostationary orbit (with a few hours' delay). X-ray flares often generate Coronal
Mass Ejections (CME) which are huge, massive bubble-like structures in the solar wind. It
takes typically 1-2 days for a CME to propagate from its origin region to the Earth distance.



The brightest and strongest auroras and disturbances in the geomagnetic field are typically
caused by CMEs. The term „geoefficiency" is used to characterize the capability of a structure
to generate variations in the near-Earth space. Besides solar wind speed and density also the
magnetic field topology of the structure is a critical factor controlling geoefficiency.
Structures whose magnetic field points in the opposite direction to Earth's magnetic field at
dayside magnetopause are particularly good in generating beautiful and extensive auroras.
Reliable information about the magnetic topology can be achieved only by in situ
measurements. For this purpose continuous solar wind measurements have been conducted at
the Langrange 1 point (L1) 1.5 million km from Earth at the Sun-Earth line since 1980's. A
typical CME propagation time from L1 to Earth is one hour, which is - with our current
scientific knowledge - also the upper limit for the lead time of reliable auroral forecasts.
Several space weather monitoring and predicting services publish alerts on X-ray flares and
earthward directed CMEs (see e.g. the service of Space Weather Prediction Center of the  US
National Oceanic and Atmospheric Administration (NOAA), http://www.swpc.noaa.gov/ and
the Solar Influences Data Center service in Belgia, http://sidc.oma.be/). Near-Real-Time
(NRT) information about geostationary energetic particle fluxes and global magnetic activity
is also available for public use. These services thus provide useful background information for
the attempts to monitor and forecast regional auroral occurrence rates.
Observations of auroral ionospheric phenomena were started in Sodankylä already during the
First Polar year 1882-83 (Seppinen and Pellinen, 2009). The Sodankylä Geophysical
observatory was established in 1913 by Finnish Academy of Science and Letters (Sucksdorff
et al., 2001). The Finnish Meteorological started regular auroral observations in Sodankylä
and in some other sites in Lapland during the International Geophysical Year (1957-58).  In
1975 Finland became a member of the scientific EISCAT association which built and started
to operate a system of incoherent scatter radars with antennas in Tromsö, Kiruna and
Sodankylä. This triggered space research groups in Sodankylä Geophysical Observatory,
Oulu University and Finnish Meteorological institute to start a collaboration in order to
conduct systematic ionospheric observations with versatile instrumentation in the
surroundings of the EISCAT radars.  Today's heritage from these activities is the MIRACLE
network of magnetometers and auroral cameras, whose data records have been used in several
studies on statistical auroral occurrence rates (Nevanlinna and Pulkkinen 2001; Partamies et



al., 2015) and on ionospheric electrodynamics linking auroras with ionosheric electric
currents and geomagnetic variations (Amm et al., 2005).
In this paper we describe a concept for an auroral forecast service (hereafter called Regional
Auroral Forecast, RAF), which is based on archived NOAA space weather alerts and regional
magnetic field and auroral recordings. The archives are used to create a set of conditional
probabilities, which tell the service user when the probability to see auroras exceeds the
average conditions in Fennoscandia during the coming 0-12 hours. The data archives and
methodology used in the development of RAF are described in Section 2. Results and a case
study on the service performance are presented in Section 3. Concluding remarks and future
prospects are discussed in Section 4.
## 2    Data and methodologies
### 2.1    Magnetic field data and their connection with auroral activity
Auroral activity is associated with variations in the geomagnetic field. During strong
geomagnetic storms the amplitude of these variations can be even 4-5 % (2000 nT) of the
strength of the main field in the Fennoscandian area (roughly 50000 nT). Typical time scales
of the disturbances vary from days (duration of a storm; Gonzalez et al., 1994) to a few
seconds (magnetic pulsations; Fukunishi et al., 1981). Magnetic variations are coupled with
visible auroras: Electron precipitation, which causes the auroral emissions by collisions with
atmospheric particles, enhances also the conductivity and electric currents in the ionosphere.
The ionospheric current system - according to the Biot-Savart law - generates magnetic
perturbations which are measurable with ground-based magnetometers.
An easy way to characterize the intensity of space weather variations is to use a proxy, which
describes the strength of ionospheric and magnetospheric currents and is based on
measurements by a global and/or local network of ground-based magnetometers.  The global
Kp-index  is one of the most widely used proxies in this area. It is defined to be the mean
value of the disturbance levels in the horizontal magnetic field component observed at 13
selected, sub-auroral stations (Bartels et al., 1939). The index has 3-hour time resolution and
its value is given in a range 0-9 according to a station specific, quasi-logarithmic scale. While
Kp describes nicely the overall space weather activity, observations of the local magnetic
field time derivative ($d\mathbf{B}/dt$) with high time resolution are a more useful way to support
regional auroral monitoring services. This linkage is utilized in an already existing public



auroral monitoring system AurorasNow! (http://aurora.fmi.fi), which was designed as a Space
Weather Applications Pilot Project with some support of the European Space Agency (ESA)
in early 2000. The service has become popular with thousands of daily visitors during winter
time.
The AurorasNow! service is based on NRT data from the Magnetometers- Ionospheric
Radars- Allsky Cameras Large Experiment (MIRACLE) network of auroral cameras and
magnetometers (http://space.fmi.fi/MIRACLE, c.f. Figure 1 and Table 1). In the original
version of Auroras Now! d**B**/dt-values from two observatories, Nurmijärvi (NUR, sub-auroral
latitudes) and Sodankylä (SOD, auroral latitudes) were monitored continuously. Enhanced
opportunity to see auroras is empirically defined to take place when the hourly maximum of
d**B**/dt exceeds 0.3 nT/s in Nurmijärvi and 0.5 nT/s in Sodankylä. More exactly, the hourly
maxima of time derivatives of X- and Y-components (geographic north and east components
with 1 minute time resolution) are calculated and the larger one is compared with the
threshold. The performance of Auroras Now! has been evaluated by comparing Sodankylä
auroral and magnetometer observations during the season from November 1 2003 to March
31 2004 (Mälkki et al., 2006). The analysis shows that in 85% of the cases when the d**B**/dt-
threshold was exceeded also auroras were observed and, on the other hand, no bright auroras
were observed when d**B**/dt-values stayed below the threshold.
RAF uses the same empirical rules between auroral occurrence and d**B**/dt were used in
AurorasNow! The threshold values for the magnetometer stations depend on the magnetic
latitudes and for additional stations used in RAF they are determined by linear interpolation
from the corresponding values of Nurmijärvi and Sodankylä. The RAF stations with their
coordinates and d**B**/dt threshold values are listed in Table 1. Stations KEV and MUO are at
latitudes poleward of the Arctic Circle (66.56°N) and under the average auroral oval during
moderate activity levels. Stations OUJ, HAN and NUR are at sub-auroral latitudes where high
d**B**/dt values are recorded only during space weather storms.
**2.2   Statistical relationship between regional magnetic field variations and**
**space weather alerts**
Forecasts of auroral activity in RAF are based on statistical relationships between space
weather alerts which describe solar and global activity and d**B**/dt values measured at the RAF
magnetometer stations. In the development work we used archives of NRT alerts by NOAA,




Halo-CME alerts by SIDC and Finnish Meteorological Institute's (FMI's) alerts for enhanced
magnetic variability based on ACE data (available in the AurorasNow! service). We
concentrate on the results based on NOAA alerts (issued 2002-2012) as they appeared to be
most useful for prediction purposes.
In the statistical analysis we sought answers to questions such as: What is the probability to
measure d$\mathbf{B}$/dt>A at station B with the Alert of type C issued T-hours earlier? Here values of
A and corresponding stations B are those listed in Table 1. The value T varies in the range 1-
48 and the different alert types (C) are described below. In practice the analysis was
conducted in the following steps:
1. Constructing a summary matrix on the NOAA alerts: Each row in the matrix
12       correspond to one hour during the years 2002-2012. Each alert type has one dedicated
13       column in the row. If that alert has been issued during the hour of the row, the variable
14       in the column is 1, otherwise it is zero.

2. Constructing a summary matrix on the hourly maxima in d$\mathbf{B}$/dt values recorded at the
16       RAF magnetometer stations. Also this matrix has values 1 (in the case of d$\mathbf{B}$/dt
17       threshold excess) or 0 (no threshold excess).

3. Determining statistical relationships between the parameters in the two matrices
19       described above: For each alert type the hours of issuance were searched and the
20       values in the d$\mathbf{B}$/dt matrix for the following 48 hours were inspected. For these 48
21       hours and for each RAF magnetometer stations the ratio W/V was determined, where
22       W is the number of hours when the threshold for auroras was exceeded and V is the
23       total number of hours in the analysis (i.e. the number of issuances of the analysed alert
24       type during the ten year period).

25    4. Identifying those NOAA alert types which yield W/V values equal to or larger than
26       0.5.

27    5. Refining the analysis of step 3 by binning the data points according to Magnetic Local
28       Time (MLT) of the RAF at issuance moment and by studying the combined effect of
29       some of the most influential alerts. Four bins were used in the local time binning:
30       Noon (06-12 UT), midnight (18-24 UT), dawn (00-06 UT) and dusk (12-18 UT).
31       (Note: for the MIRACLE local time sector Magnetic Local Time ~UT+2.5h.)
32



The NOAA archives contain the following types of alerts:
• Solar X-ray Flare alert (ALTXMF): issued when the solar X-ray flux exceeds the M5
level ($5\times10^{-5}$ W/m$^2$, at wavelengths 0.1-0.8 nm and measured at the geostationary
distances)
• Alerts on enhanced proton fluxes at the geostationary distances
(ALTPX1…ALTPX4): issued when the integral flux of protons with energies above
10 MeV exceed values 10, 100, 1000, or 10000 particle flux units (pfu).
• Alerts on enhanced electron fluxes at the geostationary distances (ALTEF3): issued
when the integral flux of electrons with energies above 10 MeV exceed value 1000
pfu.
• Solar Radio Burst alerts (ALTTP2, ALTTP4): issued in the cases of enhancements in
Type II or Type IV radio emissions with frequencies <15 MHz. Emissions are caused
by accelerated electrons in the contex of solar wind shocks and CMEs.
• Alerts on enhanced global geomagnetic activity (ALTK04…ALTK09): issued when
the Kp estimate by the Wing Kp model (Wing et al., 2005) exceeds values 4…9.
Figure 2 is an example plot on the W/V value (in %) for stations KEV and NUR during the
next 48 hours after the NOAA ALTK04 and ALTK06 issuance times. According to this plot
the probability for enhanced auroral occurrence is above 50% at KEV during ~10 hours (0
hours) after the issuance of ALTK06 (ALTK04). At the sub-auroral station NUR the
probability stays above 50% only for the first hour after the ALTK06 issuance time.
**3   Results**
**3.1   Analysis of W/V curves**
We begin the investigation of the W/V curves with the ALTXMF case, because X-ray flares
give the first signs of forthcoming space weather activity and thus they have potential to
support forecasts with longest feasible lead times. Figure 3 shows the probability curve of
ALTXMF for stations KEV, OUJ and NUR.  In this case we extend the axis of delay times up
to 120 hours in order to take into account also the impact of slowly propagating CMEs. Error
bars in Figure 3 (and in the subsequent similar figures) are determined with the standard



deviation for Poisson distribution, i.e. $\varepsilon=(\text{sqrt}(W))^{-1}(100W/V)$. ALTXMF appears not to be a
reliable enough way to forecast enhanced auroral occurence as all probability values in Figure
3 are below 50%. The impact of CMEs is visible as a moderate increase in W/V values (~15
%-units) for delay times 37-80 hours in the curves of sub-auroral stations OUJ and NUR,
where the average level of magnetic variability is low.  At KEV the baseline level of W/V is
so high (~20-30%) that no specific CME signatures can be distinguished from the background
activity. In general, the feature of ALTXMF W/V-curves staying at values <50% can be
explained with the different propagation speeds of CMEs and two factors limiting their
geoefficiency: Not all flares generate CMEs which are directed towards the Earth and not all
CMEs have the correct magnetic topology to generate high d**B**/dt values.
The W/V curves of solar radio bursts (ALTTP2, ALTTP4) and those for energetic proton and
electron enhancements (ALTEF3, ALTPX1…ALTPX4) gave similar results as those of
ALTXMF (no values exceeding 50%). The alerts on global geomagnetic activity
(ALTK04…ALTK09), however, yielded more promising results. As explained in Section 2.2,
further improvement is achieved by binning the alerts according to their issuance times. The
response at RAF stations depends on their local time sector. High W/V values are achieved
for those delay times which correspond to the situation where RAF stations are around the
midnight. UT-binning was applied only for ALTK04-ALTK06, for ALTK07 the total amount
of alerts is too small to allow MLT-binning for meaninful statistical analysis. Also for
ALTK08 and ALTK09 we still need longer data archives before any V/W curves can be
derived, but on the other hand, the curves of ALTK07 already can give a relatively good
picture of the case of exceptionally strong space weather storms and thus in the operational
RAF service probability curves from the combined ALTK07, ALTK08 and ALTK09 are
used.
Figures 4 and 5 show the W/V curves of MUO and HAN for ALTK04…ALTK06 (for the
night bin) and for ALTK07 (all points). The W/V curves of MUO and KEV are mainly
similar (latter not shown) and they describe the d**B**/dt activity at auroral latitudes: The
threshold of 50% is exceeded already after ALTK04 although only for the first hour. In the
case of ALTK05 occurence of high d**B**/dt values lasts some 7 hours after the alert and for
ALTK06 high d**B**/dt values were recorded with 50% probability for the delay hours 1-3 and
26-30. After ALTK07 enhanced d**B**/dt activity lasts some 26 hours. The W/V curves of HAN
have the same features as those of NUR (not shown). At the sub-auroral latitudes occurence




rates of high d**B**/dt values with auroral occurrence probablity >50% appear only for ALTK06
or higher and for delays of 1-11 hours. In the case of ALTK07 enhanced activity persists for
13-15 hours. The W/V curves of OUJ (not shown) are similar to those of HAN and NUR
otherwise, but the 50%-threshold of occurrence of high d**B**/dt values is exceeded already at
the activity level of ALTK05, although only during the first hour after the alert. The most
important conclusion from Figures 4 and 5 is that at auroral latitudes the occurrence rates for
high d**B**/dt are close to 50% still during the next night after the issuance of ALTK06 or
ALTK07, while at the sub-auroral stations the W/V values drop below 50% already after a
delay of 12-16 hours.
Figure 6 demonstrates the effect of UT-binning in W/V curves for MUO after ALTK06.
Again, similar behaviour appears in the W/V curves of KEV. The curves of night and dusk
sector issuance times suggest that also for the coming night V/W values are well above 50%,
while in the dawn sector issuances the on-going night is clearly more favourable for auroral
spotting than the following night. In other words, if there is already high magnetic activity in
the beginning of the dark time, it will likely continue during the nearest night hours. On the
other hand, high morning activity does not strongly indicate that the next night ~12 hours later
will still show auroral displays.

## 3.2    Description of the operational RAF service

The RAF service has been developed with ESA funding in the space weather segment of
ESA's Space Situational Awareness programme during years 2013-2015. The service has two
parts, the nowcast service which characterizes prevailing auroral occurrence probability with
the same approach as Auroras Now!, and the forecast service which uses the above described
RAF approach. In both parts the regions of enhanced auroral occurrence probabilities are
shown as bands of cyan (W/V>50%) or green (W/V>70%) color overlaid on the
Fennoscandian map. These bands are positioned at the latitudes of +/-2 degrees around the
RAF stations where the forecast d**B**/dt exceeds the threshold of enhanced probability for
auroral occurrence. The forecast service checks the latest NOAA alerts every 15 min. If alerts
of the correct type (ALTK04-09, ALTPX) have been issued during the previous 15 min the
service checks the corresponding W/V-curves for delays of T0+3, T0+6, T0+9 and T0+12
(where T0 is the alert issuance hour) and draws the forecast maps accordingly.



Figure 7 presents an example of RAF performance on the evening of Sep 07 2015. On that
day Kp values started to increase after noon so that the values for the 3-hour periods ending at
UT-times 15, 18, 21, and 24 were 4.67, 6.33, 5.67, and 6.33, respectively. The first maps
promising auroral activity appeared to the RAF service at 15:17 UT (at 18:17 local time). The
maps for T0+3, T0+6, and T0+9 (i.e. until 00:17 UT) showed bands of cyan color above KEV
and MUO stations (c.f. panel a in Figure 7). Roughly two hours later at 17:02 UT, RAF made
a radical correction in its forecasts: the forecast maps promised auroras to all latitudes for all
lead times (T0+3…T0+12), and even with >70% probability for latitudes above KEV, MUO
and OUJ until 02:02 UT (c.f. panel b in Figure 7). This time the correction was successful:
beautiful auroras were observed at several sites all over Finland. The photograph archives
maintained    by    the    Finnish    Ursa    Association    of    amateur    astronomers
(http://www.taivaanvahti.fi/observations/browse/list/1120892/observation_start_time).
contain photos on auroral displays until 00:30 UT (03:30 local time) on Sep 08 2015. The
auroral camera of MIRACLE network in Sodankylä also captured spectacular auroras for
several hours during that night (panel c in Figure 7).
Test versions of RAF have been operated at the servers of ESA and the Finnish
Meteorological Institute since May 2014. Validation studies with auroral observations from
the Ursa service and by auroral cameras of Japanese and Finnish research groups have
revealed that the performance of RAF is on a satisfactory level the case of strong, extensive
auroras (activity also at sub-latitudes), but it can miss auroral displays occurring at high-
latitudes during moderate activity. The W/V curves of KEV in Figures 2 and 3 help in
understanding this result. In both Figures the baseline level of high d$\mathbf{B}$/dt occurrence rate, i.e.
the level where W/V values settle at long delay times, is around 20-30% for KEV. This means
that at auroral latitudes nice auroral displays can take place relatively often, although no
significant global activity is ongoing. Giving case-by-case forecasts of such displays is
challenging since they most likely manifest the stochastic part of solar wind-geospace
interactions related with turbulence in the solar wind (Pulkkinen et al. 2006). It is anyway
possible to estimate the locations of the average auroral oval boundaries with statistical oval
models. Sigernes et al., (2011) present a method for deriving the oval location for different
Kp-levels. The method is based on oval models derived from optical and particle precipitation
data (Starkov 1994; Zhang and Paxton, 2008). We have compared the oval locations by the
d$\mathbf{B}$/dt approach used in RAF to those by the Starkov-oval with data from a test period (May 5-
Oct 28 2014). This comparison study suggests that these two approaches complement each
other nicely: The tool by Sigernes et al., guides users to appropriate latitudes during moderate
activity, while RAF gives a more realistic representation on oval dynamics during strong Kp
activity.

## 4   Concluding remarks and future prospects

We have used the connection between auroral sightings and rapid geomagnetic field
variations in the development of the Regional Auroral Forecast (RAF) service. The service is
based on statistical relationships between NRT alerts issued by the NOAA Space Weather
Prediction Center and d$\mathbf{B}$/dt values measured by five MIRACLE magnetometer stations
located in Finland at auroral and sub-auroral latitudes. Our data base contains NOAA alerts
and d$\mathbf{B}$/dt observations from the years 2002-2012. Magnetometer data have been used instead
of direct auroral observations when constructing the statistics, because processing numerical
data is simpler than recognizing auroras from images, whose quality can occasionally suffer
from cloudiness and moonlight contamination.
Our statistical analyses reveal that NOAA alerts on X-ray bursts or on energetic particle flux
enhancements cannot be used in the forecasts, if only probability values above 50% for
successful auroral spotting are used in the service. However, NOAA alerts on global
geomagnetic storms (characterized with Kp values >4) enable probability estimates of >50%
with lead times of 1-12 hours. RAF forecasts thus rely heavily on the well-known fact that
bright auroras appear during geomagnetic storms. The additional new piece of information
which RAF brings to the previous picture is the knowledge on typical storm durations at
different latitudes. For example, the service users southward of the Arctic Circle will learn
that after a NOAA ALTK06 issuance, auroral spotting should be done within 12 hours after
the alert, while at higher latitudes conditions can remain favourable still during the next night.
We have handled the different NOAA alert types as separate independent cases, which is a
limitation to be overcome in future studies with longer records of NOAA alerts. It is very
likely that sequences of several subsequent Kp alerts or their combinations e.g. with alerts on
enhanced energetic particle fluxes produce different probability curves for high d$\mathbf{B}$/dt values
than single alerts. The probability curves of Figure 8 support this anticipation: The
probabilities for the special case, where ALTK06 has been preceded (within 24 hours) by an
alert on enhanced proton fluxes (ALTPX*), are larger than those for the case of all ALTK06
alerts. This feature is taken into account in RAF, but obviously accounting also other alert



combinations would improve the performance of the service as soon as enough archived alert
data have been accumulated to test this hypothesis.
The threshold values which we use for d$\mathbf{B}$/dt as an implication of enhanced auroral activity
may be adjusted in the future, when we have gathered more experience in aurora data analysis
with advanced machine learning methods (Rao et al., 2014; Syrjäsuo and Partamies, 2011).
Finding optimal values for automatic recognition may, however, appear to be challenging,
since there is some variability in the user requirements (photographing versus naked eye
observations). The threshold values used in RAF come as legacy from the  Auroras Now!
service, which was designed during the years 2003-2005. These thresholds usually deserve
their place as the first approximation, but as nowadays the user community includes more
auroral photographers with high-end camera equipment than ten years ago, the detection
threshold values may need some lowering in the future RAF upgradings. Long, homogeneous
and validated records of ionospheric observations, like provided by the Sodankylä research
station and the surrounding MIRACLE network, will be crucial input for such upgrading
work.
**Acknowledgements**
The authors thank the Space Weather Prediction Center of NOAA and SIDC for providing
access to their archived space weather alerts.
The MIRACLE network is operated as an international collaboration under the leadership of
the Finnish Meteorological Institute. The IMAGE magnetometer data are collected as a joint
European collaboration. INAF-IAPS (Italy) and the University of Oulu (Finland) maintain the
ITACA ASCs and the ASC in Sodankylä.  National Institute on Polar Research (Japan) is
acknowledged for their service of auroral images which has been used in RAF testing.
A. Ketola, L. Häkkinen, S. Mäkinen, P. Posio, K. Pajunpää and A. Koistinen (all in FMI
Observation Unit) are acknowledged for their persistent and professional work for MIRACLE
observations. P. Janhunen (FMI) gave valuable advice in the analysis of W/V curves.



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



2    Table 1. Magnetometer stations used in the Auroras Now! and RAF services and the

3    corresponding d**B**/dt threshold for enhanced probability of aurora occurrence. Magnetic

4    latitude (MLAT) is given in the frame of Corrected Geomagnetic Coordinates.

| Code | Name | Geogr. Coord. | MLAT | dB/dt Threshold |
|------|------|---------------|------|-----------------|
| NUR | Nurmijärvi | 60.50°N, 24.65°E | 56.9 | 0.30 nT/s |
| HAN | Hankasalmi | 62.25°N, 26.60°E | 58.7 | 0.35 nT/s |
| OUJ | Oulujärvi | 64.52°N, 27.23°E | 61.0 | 0.42 nT/s |
| SOD | Sodankylä | 67.37°N, 26.63°E | 63.9 | 0.50 nT/s |
| MUO | Muonio | 68.02°N, 23.53°E | 64.7 | 0.52 nT/s |
| KEV | Kevo | 69.76°N, 27.01°E | 66.3 | 0.57 nT/s |



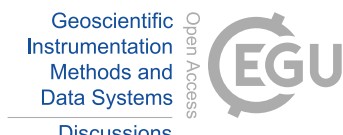

2 Table 2. Amounts of NOAA alerts used in the study. For Kp=4-8 the four values in the given

3 sums are amounts of the events which took place in the local time bins of dawn, dusk, night

4 and noon (for more details see text).

| Alert | # |
|---|---|
| ALTEF3 | 1459 |
| ALTK04 | 350+249+129+267=995 |
| ALTK05 | 177+92+54+126=449 |
| ALTK06 | 71+40+21+63=195 |
| ALTK07 | 16+11+12+20=59 |
| ALTK08 | 2+2+6+7=17 |
| ALTK09 | 5 |
| ALTPC0 | 31 |
| ALTPX1 | 92 |
| ALTPX2 | 43 |
| ALTPX3 | 19 |
| ALTPX4 | 1 |
| ALTTP2 | 377 |
| ALTTP4 | 196 |
| ALTXMF | 159 |



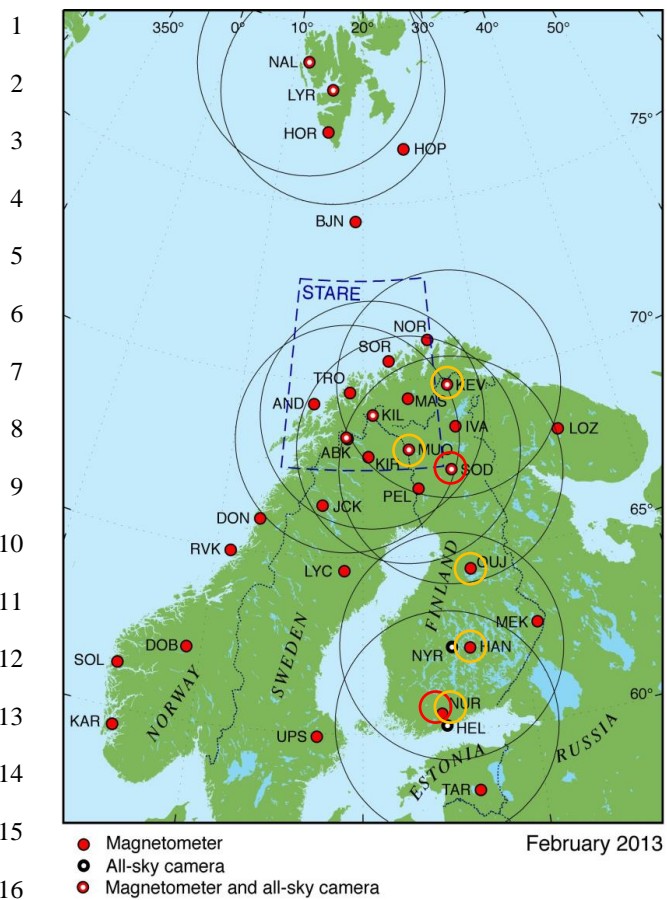

Figure 1. Stations of the MIRACLE network. The field-of-views of auroral cameras are shown with black circles and the observing area of the bi-static STARE radar (operated 1997-2005) with the rectangle (dashed lines). Magnetometer stations used in the RAF and Auroras Now! services are show with the small red and orange circles, respectively.





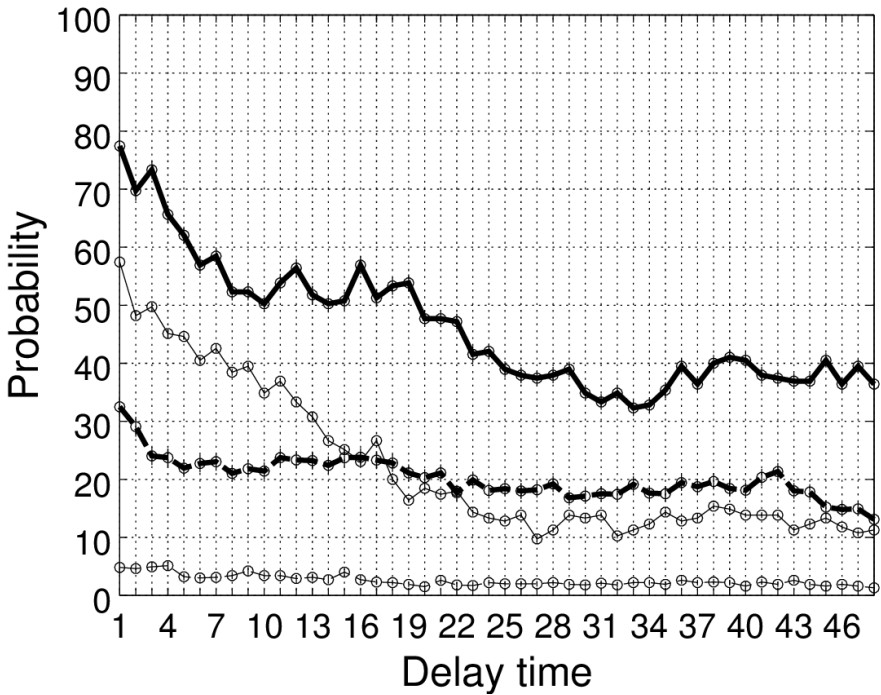

Figure 2. W/V values (in %) for stations KEV (thick lines) and NUR (thin lines) during 48
hours after the issuance of ALTK06 (solid lines) and ALTK04 (dashed lines). W is the
number of cases with d$\mathbf{B}$/dt excess above the threshold for enhanced auroral occurrence. V is
number of ALTK06 (195) and ALTK04 (995) issued during the years 2002-2012.





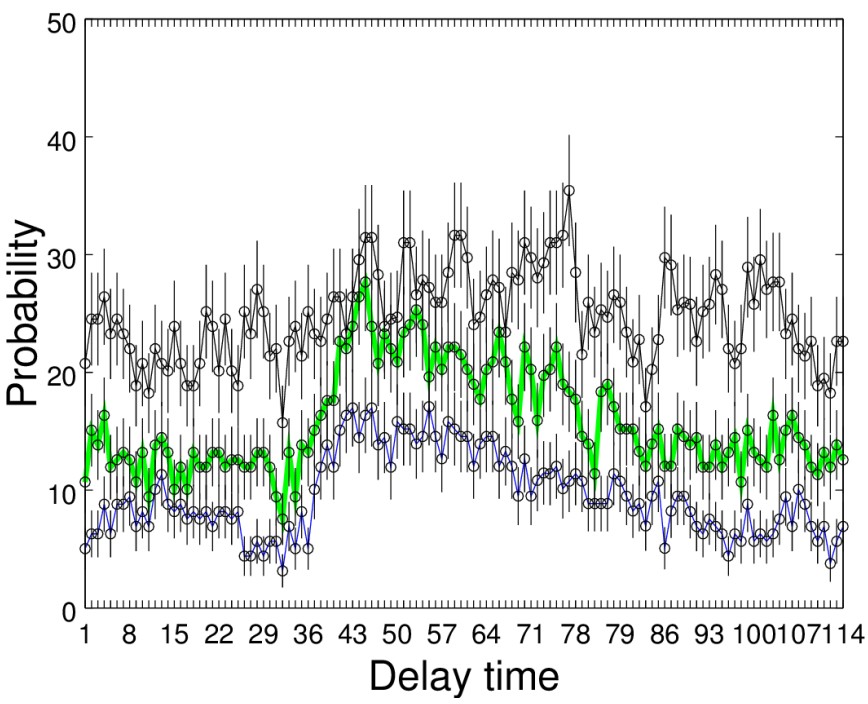

Figure 3. W/V values (in %) for stations KEV (black), OUJ (green) and NUR (blue) during
120 hours after the issuance of ALTXMF. W is the number of cases with d**B**/dt excess above
the threshold for enhanced auroral occurrence. V is number of ALTXMF (159) issued during
the years 2002-2012.





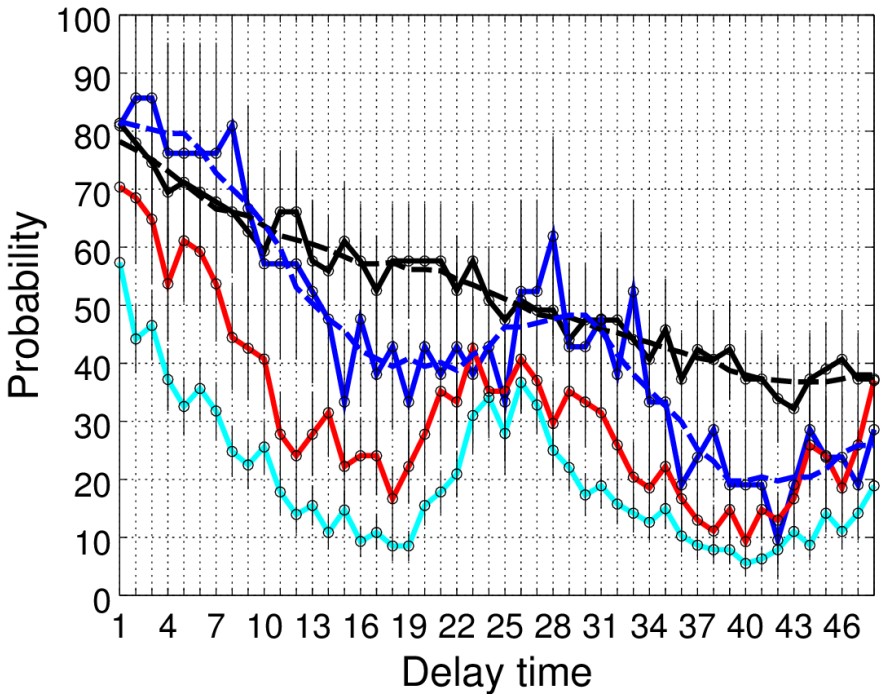

Figure 4. W/V values (in %) of station MUO for ALTK04 (cyan), ALTK05 (red), ALTK06
(blue) and for ALTK07 ((black). The curve for ALTK07 is based on all data points, while for
the other activity levels only the points of night time bin has been used (for the amounts of
data points, see Table 1). The dashed lines represent smoothed curves (7-point running
averages) for ALTK06 and ALTK07, which are used in the operational RAF service.





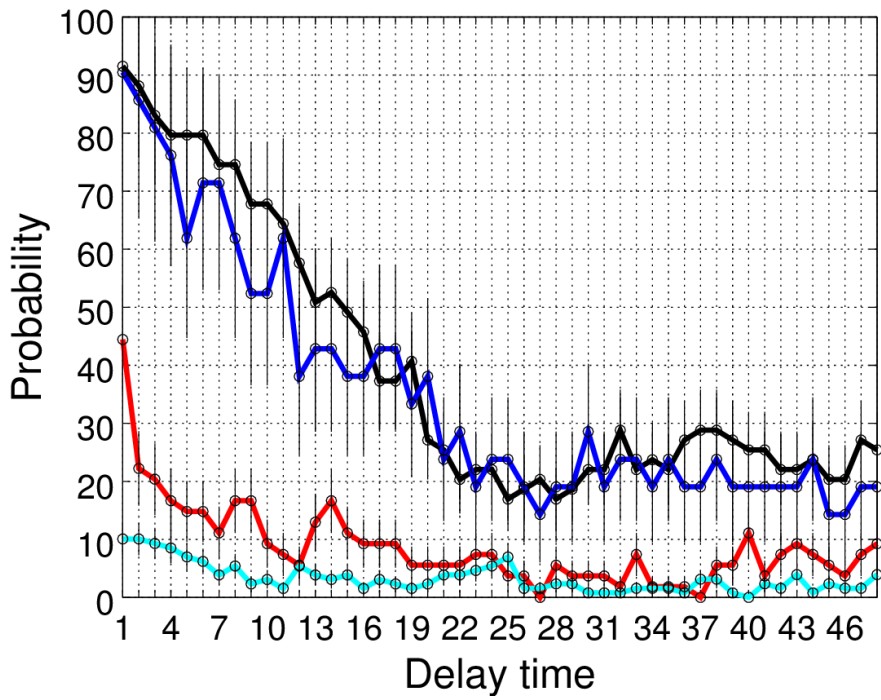

Figure 5. W/V values (in %) of station HAN for ALTK04 (cyan), ALTK05 (red), ALTK06
(blue) and for ALTK07 ((black). The curve for ALTK07 is based on all data points, while for
the other activity levels only the points of night time bin has been used (for the amounts of
data points, see Table 1).

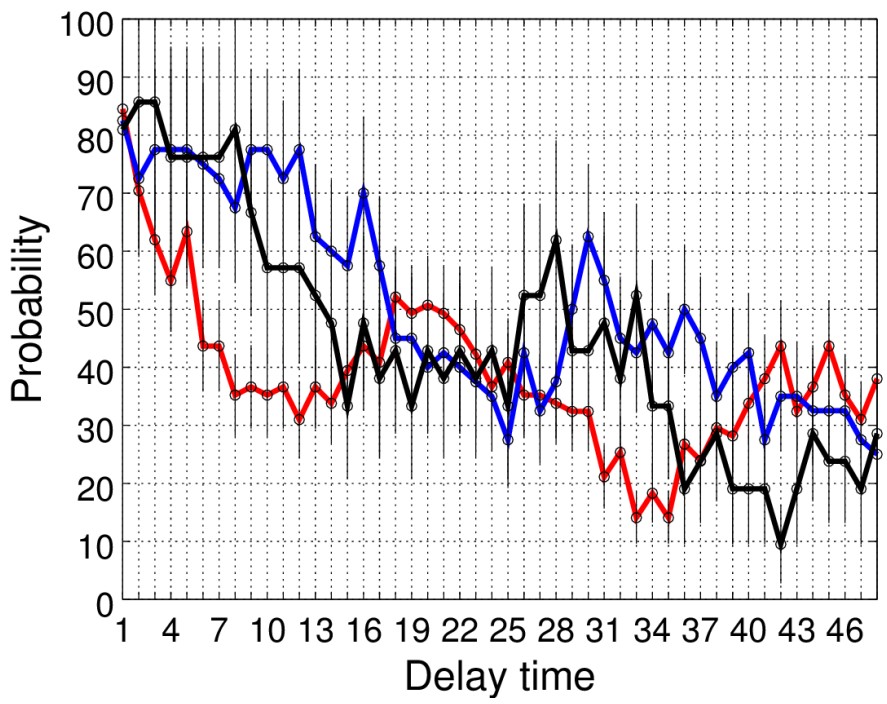

Figure 6. W/V values (in %) of station MUO for ALTK06 and the UT bins of dawn (red), dusk (blue) and night (black).

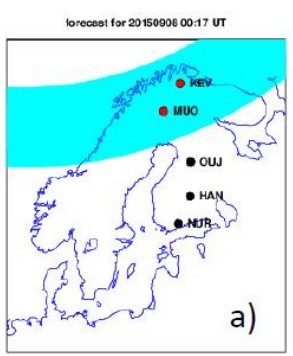
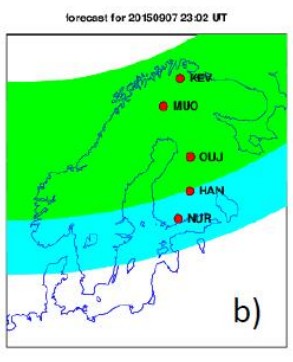
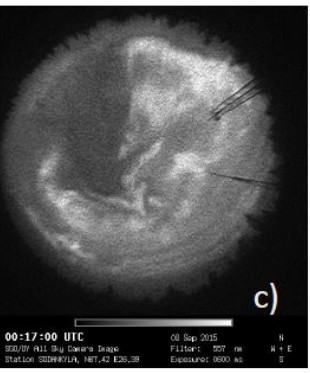

Figure 7. RAF forecasts on auroral occurrence probability for a couple of time instants around the midnight on Sep 07-08 2015 and an example image from the Sodankylä auroral camera station from the same time period. The forecasts were published at (a) 15:17 UT and (b) 17:02 UT. Cyan (green) color gives regions with >50% (>70%) probability of auroral sightings.





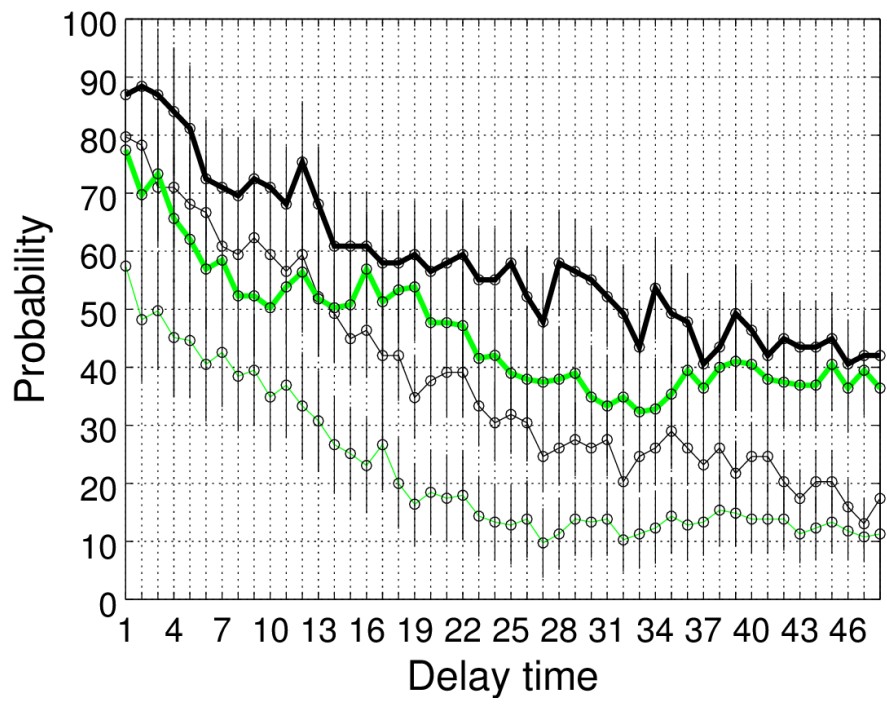

2    Figure 8. W/V values (in %) of station KEV (black) and NUR (Green) for the special case of

3    ALTPX* preceding ALTK06 (thick lines) and for the case of all ALTK06 events. The

4    number of data points in the bin of special cases is 69.

