# Peer review of "Forecasting auroras from regional and global magnetic"

_Geoscientific Instrumentation, Methods and Data Systems, 2015_

## Referee Comment (RC1) · Anonymous Referee #1 · 20 Feb 2016

General comment.

The paper describes a concept for a regional auroral forecast (RAF) service intended for users interested in viewing the Aurora. The main emphasis is given to to Finland, but presumably the service may be expanded to include other geographic areas as well. The service, which is statistical in nature, is based upon archives of ground based magnetic field recordings and auroral observations as well as space weather alerts provided by NOAA, most successfully alerts based on Kp forecasts (Wing). Forecasts up to 12 hours ahead are successfully given, with extended time range for high latitudes.

The paper is well written and contains the necessary descriptions and background for a scientific paper. It also contains the sufficient amount of new findings. Any major problems with the paper has not been identified, therefore publication is recommended.

Below, some minor comments are given, based on aspects that came to mind while reading

Special comments

As is often occurring in context of space weather related studies, emphasis is given to elevated conditions ranging from minor to severe geomagnetic storms. Some discussion should be added to the fact that in the auroral zone proper, which may be considered north of main-land Fennoscandia (Barents sea), the occurence of non storm-time substorms is very common, and thus bright aurora might be expected for short time intervals under ambient solar wind conditions. As is pointed out on page 5, line 24, the northern stations used in the study are under the auroral oval under somewhat elevated activity levels ("moderate activity levels"). Although, these stations are located almost as north as aurora viewing is possible in this time sector, for the sake of applicability to other sectors such as the North American, some discussion could be added. Furthermore, the lack of NOAA alerts during quiet time substorms should be addressed.

Regarding the issuing of Kp alerts from NOAA, a few sentences discussing what problems the Wing-model might introduce, wrt. accurancy, should be added.

On page 10 there is a brief description of the auroral oval predictions performed by Sigernes et al. Mention of other techniques such as the Ovation Prime by NOAA may be considered. Furthermore, although not critical, alternative methods or proxies for determining the location of the auroral oval such as Johnsen (2013) (http://www.swsc-journal.org/articles/swsc/abs/2013/01/swsc130002/swsc130002.html) and references therein, where the eastward and westward electrojets are used as a proxy for the auroral oval, may also be mentioned.

Minor comment.

Page 5 line 3 Do you mean early 2000s or January 2000?

---

## Short Comment (SC1) · 11 Mar 2016

Reply to Referee #1 concerning GI-manuscript gi-2015-33 ("Forecasting auroras from regional..." by Kauristie et al.,)

We thank the Referee for constructive comments. Below are our replies and suggestions for some modifications in the manuscript:

Predicting auroras during ambient solar wind conditions:

Like the Referee correctly points out, our approach is not useful in the attempts to forecast auroras at high latitudes during non-storm times. Statistical analysis of FMI all-sky camera data shows that during the best years of auroral activity (some 2-3 years after sunspot maxima) the occurrence rates are 60-75% at stations under the auroral oval

(i.e. at magnetic latitudes 64-75, stations SOD and LYR, in Fig.1 below). Comparing these values to the threshold which we use in RAF for enhanced auroral activity (occurrence probability >50%) reveals that cloudiness forecasts provide at auroral latitudes more useful information for auroral spotting than RAF statistics. At sub-auroral latitudes an announcement of enhanced probability by RAF can be interpreted to represent conditions which prevail at auroral latitudes during the most favorable years in solar cycle. With the latitudinal coverage of FMI all-sky and magnetometer observations we conclude that the auroral oval latitudes in this context correspond roughly to magnetic latitudes 64-75, while latitudes below MLAT 61 represent clearly sub-auroral regions.

An older version of the Fig. 1 (attached) has been presented by Pulkkinen et al. (2011). We suggest to add the updated version of this Figure to our manuscript with some discussion along the lines given above.

Potential problems due to usage of Wing model for Kp:

Bala and Reiff (2014) have tested the performance of Wing Kp 1-hour forecast with real-time output values collected during a test period of 22 months (April 2011 – February 2013). This study shows that the Wing Kp approach has some tendency to overestimate Kp values during enhanced activity. In the test data set of 15960 time instants the Wing approach claimed the Kp to be equal or more than 4 in 1222 cases. Checking against the official Kp values reveals that 335 of these were false alarms (i.e. the real Kp was <4). We will add this remark to the next version of our manuscript.

Additional references:

Yes, certainly the NOAA Ovation service and the identification method of auroral electrojet boundaries by Johnsen (2013) should be mentioned in our discussion. Minor comment: We mean early 2000s.

References: Bala, R., and P. Reiff, Validating the Rice neural network and the Wing Kp

real-time models, Space Weather, 12, 417-425, doi:10.1002/2014SW001075, 2014.

Johnsen, M.G., Real-time determination and monitoring of the auroral electrojet boundaries, J. Space Weather Space Clim., 3, A28, doi:10.1051/swsc/2013050, 2013.

Pulkkinen, T.I., E.I. Tanskanen, A. Viljanen, N. Partamies and K. Kauristie, Auroral electrojets during deep solar minimum at the end of solar cycle 23, J. Geophys. Res., 116, A04207, doi:10.1029/2010JA016098, 2011.

Please also note the supplement to this comment:
http://www.geosci-instrum-method-data-syst-discuss.net/gi-2015-33/gi-2015-33-SC1-supplement.pdf

———————————————————

**HAN** **SOD** **KIL** **LYR**

**Fig. 1.** Figure 1: Occurrence probability of auroras at FMI stations with the following magnetic latitudes: HAN 59; SOD 64; KIL 66; LYR 75. Probabilities are based on visual inspections of quick look data (keo

---

## Referee Comment (RC2) · Anonymous Referee #2 · 22 Mar 2016

GENERAL COMMENTS

This manuscript presents a method to forecast or nowcast aurora. The forecast method is based on global geomagnetic activity alerts issued by NOAA, which are statistically correlated to expected regional magnetic activity levels during the following 0-48 hours and thereof to occurrence probability of visual aurora.

At first glance the proposed forecast method seems to rely on duct tape and haywire, as it builds several layers of semi-empirical models and alert services on top of each other. However, a closer inspection reveals that the forecast method is well defined and quite logical. Also the technical implementation is relatively straightforward, once the statistical correlations and other empirical relationships have been obtained.

The manuscript is written in a clear and logical manner. The forecast method is ade-

quately explained and the presented results indicate that it works reasonably well.

In summary, this is an interesting paper, and is suitable for publication in EPS after some relatively minor modifications.

SPECIFIC COMMENTS

P1 L18-23: This explanation makes the abstract bit unclear, first you say RAF use one dataset, then you say that it's based on some other datasets. Please clarify this (I think P11 L6-9 could be good start).

P2 L13-18: You should make it more clear that this is a direct quote from Lilensten et al.

P5 L17: The definition of "bright" may be somewhat subjective, but where you base the assessment "no bright auroras were observed"? I did not read the report completely, but in their section 6.1, tables 1 and 2, Malkki et al. indicate that in about 13-18% of the cases "the system missed clear auroras".

P5 L21: In addition to interpolation, also extrapolation is needed for the northernmost stations. You should at least mention if not properly discuss the reliability of extrapolation from a marginally auroral station (SOD) to auroral stations (KEV, MUO). It's clear that the linear extrapolation must break down somewhere.

P6 point 3: How you handle cases where alerts are separated by less than 48 hours? Has this some effect on your analysis or results?

P9 L1, maybe also elsewhere: If I understood correctly, this is the probability of exceeding the dB/dt threshold, which may be different from the probability of observing aurora.

P9 L10-17: What about the noon curve, how well it indicates aurora during the following night?

P9 L26: You mean that the bands are positioned around the geomagnetic pole at the

magnetic latitudes +-2 deg...?

P9 L30: I assume that the curves obtained by UT binning are used?

TECHNICAL COMMENTS

P3 L2: For typing the opening and closing quotation marks in English, see https://en.wikipedia.org/wiki/Quotation_marks_in_English or chapter 2.4.1 in the not so short introduction to latex (http://www.ctan.org/tex-archive/info/lshort/english/). This also applies to the apostrophes in hours' (P2 L29) and 1980's (P3 L9).

P3 L2: You mean "solar wind structure"?

P5 L3: Do you mean the year 2000 or the decades 2000's?

P9 L29: Inclusion of ALTPX appears here out of nowhere and is explained only much later.

P10 L4+7: I believe many forecasters avoid the word "promise", but that's up to you.

P10 L19: level in the case

P12 L6: What you mean by "may appear to be challenging"? You do not yet know how it appears to be?

NON-SERIOUS COMMENT

P11 L12: Your offhand remark, that auroral images can occasionally suffer from cloudiness and moonlight, reminds me of an anecdote about the book Principia Mathematica by Whitehead and Russell. They set out to derive all of mathematics from the axioms of formal logic, and when they at page 379 prove that 1+1=2, they remark that this result is occasionally useful.

---

## Author Comment (AC1) · 29 Apr 2016

We thank the Referee for the valuable comments to improve our paper. Below we give our replies and suggestions for the modifications in the manuscript:

*P1 L18-23, clarifying the Abstract:

We agree, the first sentences in the Abstract are confusing. According to the recommendation of the Referee we suggest the following reformulation:

We use the connection between auroral sightings and rapid geomagnetic field variations in a concept for a Regional Auroral Forecast (RAF) service. The service is based on statistical relationships between near-real-time alerts issued by the NOAA Space Weather Prediction Center and magnetic time derivative ($dB/dt$) values measured by

five MIRACLE magnetometer stations located in Finland at auroral and sub-auroral latitudes. Our data base contains NOAA alerts and dB/dt observations from the years 2002-2012. These data are used to create a set of conditional probabilities, which. . .

*P2 L13-18, direct quote from Lilensten et al

We suggest the following modification:

According to Lilensten et al. (2008) "Space weather is the physical and phenomenological state of natural space environments. The associated discipline aims, through observations, monitoring, analysis and modeling, at understanding and predicting the state of the Sun, the interplanetary and planetary environments, and the solar and non-solar driven perturbations that affect them, and also at forecasting and nowcasting the potential impacts on biological and technological systems." Auroras are harmless. . .

*P5 L17, definition of bright auroras

We have to admit that we are a bit on a slippery ground with this statement. Unfortunately we do not have exact intensity calibration information for the cameras which were used in the validation of Auroras Now as reported by Mälkki et al. (2006). We know, however, that on those days (when the image intensifier of the camera was still new), the system was somewhat more sensitive than the human eye. So our optimistic conclusion in the report by Mälkki et al. was that in the 13% of cases when Anow failed the auroras were dim and thus not of interest for auroral tourism. Like explained in the concluding remarks (P12 L9-12), nowadays the situation is more complicated as optimally RAF should be able to serve both photographers with very sensitive cameras and observing with naked eye.

We suggest the following modification:

The analysis shows that in 86% of the cases when the dB/dt-threshold was exceeded also auroras were observed. In the 13% of the cases when Auroras Now! failed to spot the auroras, the intensities were typically dim or even below the sensitivity of human

eye.

*P5 L21, extrapolation of the threshold value

We suggest the following modification:

...in RAF they are determined by linear inter- and extrapolation from the corresponding values of Nurmijärvi and Sodankylä. The statistical study of Finnish all-sky camera recordings from years 1973-1997 by Nevanlinna and Pulkkinen (2001) shows that assuming a linear trend in the auroral occurrence probability according latitude is a good approximation at magnetic latitudes $63°$-$70°$. At latitudes below $63°$ the evidence for a linear trend is less clear, but as all-sky observations from these latitudes are scarce in the analyzed data base, we use the linear relationship also there as the first approximation. The RAF stations with...

*P6 point 3, handling of alerts with less than 48 hour separation

Just for simplicity reasons each alert was handled as an individual case (i.e. alerts with less than 48 hour separation were handled similarly as the other alerts). Obviously improving this part of our approach would be a good topic for a future upgrading of the service with a larger NOAA data set.

We suggest the following modification:

...W/V was determined, where W is the number of hours when the threshold for auroras was exceeded and V is the total number of hours in the analysis (i.e. the number of issuances of the analysed alert type during the ten year period). The combined effect of subsequent alerts was ignored in the analysis as alerts with less than 48 hour separation were handled as independent separate cases.

*P9 L1, probability of exceeding the dB/dt threshold versus probability of auroras

We suggest the following modification to P7 L17-20 where we introduce the W/V plots for the first time:

In the following discussion we use the W/V value (in %) as a proxy for the auroral occurrence probability, although exactly speaking this value represents the probability of dB/dt excess above the given threshold. Figure 2 is an example plot on W/V for stations KEV and NUR during the next 48 hours after the NOAA ALTK04 and ALTK06 issuance times. According to this plot the probability for enhanced auroral occurrence is above 50% at KEV during ∼10 hours (0 hours) after the issuance of ALTK06 (ALTK04).

*P9 L10-17, how well noon curves indicate aurora during the following night

A good point. ALTK06 issuance around noon indicates ≥60% auroral probabilities for the coming night.

We suggest the following modification to the text:

The curves of night and dusk sector issuance times suggest that for the coming night V/W values are well above 50%. ALTK06 issuance around noon also indicates ≥60% auroral probabilities for the coming night (curves not shown). In the case of dawn sector issuances the on-going night is clearly more favourable for auroral spotting than the following night. In other words, if there is already high magnetic activity in the beginning of the dark time, it will likely continue during the nearest night hours. On the other hand, high morning activity does not strongly indicate that the next night ∼12 hours later will still show auroral displays.

*P9 L26, location of bands with enhanced activity

No, we mean that the band envelopes with +/- 2° the magnetic latitude of the station which has measured a dB/dt excess.

We suggest the following modification to the text:

In both parts the regions of enhanced auroral occurrence probabilities are shown as bands of cyan (W/V>50%) or green (W/V>70%) color overlaid on the Fennoscandian map. These bands are positioned at the latitudes of +/-2 degrees around the RAF stations where the forecast dB/dt exceeds the threshold of enhanced probability for

auroral occurrence (for an example, see Fig 7).

*P9 L30, location of bands with enhanced activity

Yes, we use the curves based on UT-binning.

We suggest the following modification to the text:

The forecast service checks the latest NOAA alerts every 15 min. If alerts of the correct type (ALTK04-09, ALTPX) have been issued during the previous 15 min the service checks the corresponding W/V-curves with UT-binning for delays of T0+3, T0+6, T0+9 and T0+12 (where T0 is the alert issuance hour) and draws the forecast maps accordingly.

*TECHNICAL COMMENTS

P3 L3: "geoefficiency" -> geoefficiency (in italics)

P2 L29 (with a few hours' delay) -> (with a few hours' delay) (Although my Office refuses to make this change in Times font).

P3 L9 since 1980's. -> since 1980s.

P3 L2 capability of a structure -> capability of a solar wind structure

P5 L2 in early 2000. -> in early 2000s.

Inclusion of ALTPX (P9 L29): ALTPX alerts are introduced for the first time on P7 L6

P10 L4&7 promise -> forecast

P10 L19 level the case -> level in the case

P12 L6 recognition may, however, appear to be challenging, since -> recognition will not be straightforward, since

Thanks also for the non-serious comment!